# Extrapulmonary tuberculosis in HIV-infected patients in rural Tanzania: The prospective Kilombero and Ulanga antiretroviral cohort

Armon Arpagaus[1,2☯], Fabian Christoph Franzeck[3☯], George Sikalengo[4,5], Robert Ndege[4,5], Dorcas Mnzava[4], Martin Rohacek[1,2,4], Jerry Hella[1,2,4], Klaus Reither[1,2], Manuel Battegay[3], Tracy Renee Glass[1,2], Daniel Henry Paris[1,2], Farida Bani[4], Omary Ngome Rajab[4], Maja Weisser[1,3,4]*, on behalf of the KIULARCO Study Group[¶]

1 Swiss Tropical and Public Health Institute, University of Basel, Basel, Switzerland, 2 University of Basel, Basel, Switzerland, 3 Division of Infectious Diseases & Hospital Epidemiology, University Hospital Basel, University of Basel, Basel, Switzerland, 4 Ifakara Health Institute, Ifakara, United Republic of Tanzania, 5 Saint Francis Referral Hospital, Ifakara, United Republic of Tanzania

☯ These authors contributed equally to this work.
¶ Membership of the KIULARCO Study Group is provided in the Acknowledgments.
* m.weisser@unibas.ch

**Data Availability Statement:** All relevant data are within the paper and its Supporting Information files.

## Abstract

### Background

In sub-Saharan Africa, diagnosis and management of extrapulmonary tuberculosis (EPTB) in people living with HIV (PLHIV) remains a major challenge. This study aimed to characterize the epidemiology and risk factors for poor outcome of extrapulmonary tuberculosis in people living with HIV (PLHIV) in a rural setting in Tanzania.

### Methods

We included PLHIV >18 years of age enrolled into the Kilombero and Ulanga antiretroviral cohort (KIULARCO) from 2013 to 2017. We assessed the diagnosis of tuberculosis by integrating prospectively collected clinical and microbiological data. We calculated prevalence- and incidence rates and used Cox regression analysis to evaluate the association of risk factors in extrapulmonary tuberculosis (EPTB) with a combined endpoint of lost to follow-up (LTFU) and death.

### Results

We included 3,129 subjects (64.5% female) with a median age of 38 years (interquartile range [IQR] 31–46) and a median CD4+ cell count of 229/µl (IQR 94–421) at baseline. During the median follow-up of 1.25 years (IQR 0.46–2.85), 574 (18.4%) subjects were diagnosed with tuberculosis, whereof 175 (30.5%) had an extrapulmonary manifestation. Microbiological evidence by Acid-Fast-Bacillus stain (AFB-stain) or Xpert® MTB/RIF was present in 178/483 (36.9%) patients with pulmonary and in 28/175 (16.0%) of patients with extrapulmonary manifestations, respectively. Incidence density rates for pulmonary Tuberculosis (PTB and EPTB were 17.9/1000person-years (py) (95% CI 14.2–22.6) and 5.8/1000

**Funding:** This research was supported by grants from the Freie Akademische Gesellschaft Basel (https://fag-basel.ch). The funders had no role in study design, data collection and analysis, decision to publish, or preparation of the manuscript.

**Competing interests:** The authors have declared that no competing interests exist.

py (95% CI 4.0–8.5), respectively. The combined endpoint of death and LTFU was observed in 1058 (33.8%) patients, most frequently in the subgroup of EPTB (47.2%). Patients with EPTB had a higher rate of the composite outcome of death/LTFU after TB diagnosis than with PTB [HR 1.63, (1.14–2.31); p = 0.006]. The adjusted hazard ratios [HR (95% CI)] for death/LTFU in EPTB patients were significantly increased for patients aged >45 years [HR 1.95, (1.15–3.3); p = 0.013], whereas ART use was protective [HR 0.15, (0.08–0.27); p <0.001].

## Conclusions

Extrapulmonary tuberculosis was a frequent manifestation in this cohort of PLHIV. The diagnosis of EPTB in the absence of histopathology and mycobacterial culture remains challenging even with availability of Xpert® MTB/RIF. Patients with EPTB had increased rates of mortality and LTFU despite early recognition of the disease after enrollment.

## Introduction

Tuberculosis (TB) remains a leading cause of death, especially in patients co-infected with HIV. Of the 10 million patients diagnosed with TB globally in 2017, 9% were co-infected with HIV. Of these, 72% live in Africa and contribute 84% of the 300'000 deaths [1]. Tanzania is one of the thirty high-endemic countries with an estimated TB incidence rate of 269/100'000 in the general population, an HIV-TB coinfection rate of 31% and a mortality rate in HIV-co-infected patients of 39/100'000 [1].

While TB is primarily a pulmonary disease, extrapulmonary manifestations account for 14% of incident TB cases worldwide [1]. In countries with a high HIV/TB co-infection rate such as Tanzania, 20% of incident TB manifest as extrapulmonary TB (EPTB) [1]. TB can affect any organ—however, lymph nodes and pleura are the most frequent localizations [2, 3]. In African settings TB lymphadenitis has accounted for 61%–78% and TB pleurisy for 10.6% of EPTB cases [4–6].

Diagnosis of EPTB remains challenging in resource-limited settings, as clinical manifestations are diverse, microbiological testing depends on invasive procedures and the gold standard of mycobacterial culture or supporting histopathology is mostly unavailable. The low bacillary load in tissues leads to poor sensitivity of microscopy for acid-fast bacilli (AFB) [7]. In a recent metanalysis on a wide range of different samples including cerebrospinal fluid, lymph nodes and a variety of tissues of patients with suspected EPTB, Xpert® MTB/RIF showed a high specificity of ≥ 98% in cerebrospinal fluid, pleural fluid, urine, and peritoneal fluid compared to culture, whereas the sensitivity ranged from 31% in pleural tissue to> 80% in urine and bone or joint fluid and tissue and 97% in bone or joint fluid compared to culture [8].

This low sensitivity of microbiological tests and the lack of diagnostic tools for important differential diagnoses of EPTB, e.g. malignancy or heart failure in case of pleural effusions [9], result in diagnostic uncertainty. As a consequence, many patients with compatible symptoms receive anti-tuberculous treatment without confirmatory testing [10], resulting in overtreatment of tuberculosis, and underdiagnosis of alternative conditions [11, 12].

Data on outcome and predictors of survival are scarce and most knowledge comes from retrospective studies. Poor outcomes have been reported in patients with central nervous manifestations [13], lymphopenia [14], older age and late presentation [15].

Our study aimed to characterize the epidemiology of EPTB in people living with HIV (PLHIV) in a rural setting in Tanzania, and to evaluate risk factors of poor outcomes such as LTFU or death and predictors for this outcome.

## Methods

### Study site and patient population

For this prospective study, we included all patients aged ≥18 years enrolled into the Kilombero and Ulanga Antiretroviral Cohort (KIULARCO) between January 1st, 2013 until December 31st, 2017 who completed a baseline evaluation and at least one clinical follow-up. Exclusion criteria were lack of informed consent, no follow-up visits or age <18years.

The KIULARCO is an ongoing cohort of HIV-infected patients in care at the Chronic Diseases Clinic Ifakara (CDCI) located at the St. Francis Referral Hospital in the southwest of Tanzania. Since its starts 2005, KIULARCO has enrolled more than 10,000 consenting patients with approximately 4,000 patients currently under active follow-up. The CDCI provides services for HIV according to guidelines issued by the national AIDS control program (NACP). While initially treatment guidelines recommended start of cART for patients with an HIV infection WHO stage III/IV or CD4 cell count below 350cells/ul [16], the CD4 threshold was adapted to 500cells/ul in 2013 [17] and removed to allow treatment for all in 2015 [18].

Within routine care, stable patients are assessed by a clinician 4 times per year. During clinical visits, information on demographics, clinical parameters, co-morbidities, prescription of cART and co-medications, adherence and laboratory monitoring are entered into a customized electronic clinical record system designed for subsequent scientific analysis (OpenMRS, http://openmrs.org) [19, 20]. In 2012, the clinic integrated TB services and implemented systematic screening for TB with the World Health Organization (WHO) symptom screening tool and Xpert® MTB/RIF for patients with suspected TB (Cepheid, USA) [21–24]. In patients suspected of PTB, sputum smear and, in patients with EPTB, urine or a sample from a sterile site is examined by bacilloscopy and Xpert® MTB/RIF analysis. In addition, from July 2016 to June 2017, a prospective observational study on the value of sonography to rule out tuberculosis was performed at St. Francis Referral Hospital according to the Focused Assessment with Sonography for HIV and Tuberculosis protocol (FASH) [25]. The study included HIV-positive and -negative patients with symptoms of TB and comprised performance of sonography to detect pleural or cardiac effusion, tuberculoma of liver and spleen, ascites, abdominal lymphadenopathy and ileum wall thickening. Xpert®MTB/RIF testing from sputum, urine samples and sterile fluids (thoracentesis, pericardiocentesis, ascites tap) improving standards for diagnosis of TB.

### Definitions

To define PTB and EPTB, the International Classification of disease (ICD-10) [26] code and results from microbiological tests were used (S1 Table). Due to the unavailability of a local pathology service, no histopathological features were included in the disease definitions.

Microbiologically confirmed PTB was diagnosed, if a sputum sample was positive or an ICD-10 code for PTB (A15.x and A16.x except A15.6 and A16.5) was documented, and TB treatment was initiated 1 week before until 3 months after the working diagnosis of suspected TB. Microbiologically confirmed EPTB was diagnosed, if acid-fast bacilli on microscopy or Xpert® MTB/RIF were positive from an extrapulmonary site. Additionally, if an ICD-code for TB pleurisy (A15.6) was documented and anti-TB treatment was started 1 week before until 3 months after the working diagnosis, the infection was considered as confirmed. Microbiologically confirmed combined PTB and EPTB was defined as either a) presence of acid-fast bacilli/

positive Xpert MTB/RIF from sputum and A15.6, A17, A18 or A19 or b) presence of acid-fast bacilli/positive Xpert MTB/RIF from a sterile site and cough or infiltrate on a chest x-ray without an alternative explanation.

Clinical PTB was defined as ICD-10 code A16 (excluding A16.5) without microbiological confirmation and start of TB treatment 1 week before until 3 months after working diagnosis of suspected TB. Clinical EPTB was defined as ICD-10 code A16.5, A17, A18, A19 without microbiological confirmation and without cough or infiltrate on chest x-ray, and start of TB treatment 1 week before until 3 months after working diagnosis of suspected TB.

Combined clinical PTB and EPTB was defined as A16.5, A17, A18, A19 and presence of cough or infiltrate on chest x-ray without microbiological confirmation, and initiation of an anti-tuberculous treatment starting 1 week before until 3 months after an established working diagnosis.

No presence of tuberculosis was defined as absence of positive microbiological results, ICD-10 code for tuberculosis and no initiation of anti-tuberculosis therapy.

Cases of TB diagnosed during the first 90 days of observation after enrollment were judged as prevalent at enrolment to allow for delays in diagnoses in order to prevent over classification of incident cases. Group descriptions using the term "extrapulmonary manifestation" include patients in the EPTB as well as the EPTB/PTB group.

Anemia was defined as ICD-10 codes D50-D64.9 or by a hemoglobin level below 12g/l or 11g/l for men and women, respectively. Arterial hypertension was defined as ICD-10 codes I10—I15 or measured blood pressure of >140/90mmHg upon two consecutive measurements or active antihypertensive treatment. Undernutrition was defined as ICD-10 codes E44-E46 or a body mass index (BMI) <18.5kg/m$^2$, obesity as E66.0-E66.9 or a BMI >30.0kg/m$^2$. Acute kidney failure was diagnosed by ICD-10 codes N17.0–17.9, chronic kidney failure by N18.0–18.9 or an uninterrupted estimated glomerular filtration rate (according to the CKD-EPI formula) <60ml/min/1.73m$^2$ for > 3 months.

Patients were considered being lost to follow-up (LTFU) if they did not present to the clinic within 60 days after their last scheduled appointment. We decided to use a composite clinical outcome of death and LTFU due to a significant rate of LTFU in our cohort and an assumed high mortality in those patients. [27].

## Data management and analysis

All included patients were assigned to the 4 following groups: PTB, EPTB, combined PTB and EPTB and no TB. Continuous variables were summarized using median and interquartile range and categorical variables with counts and frequencies. Comparisons of continuous and categorical variables were done using the Mann–Whitney U test and the chi-square test, respectively. Prevalence and incidence density rates (i.e. count of incident cases per 1000py of subjects at risk) were calculated including confirmed and clinically diagnosed cases of tuberculosis. Kaplan–Meier estimates were used to plot cumulative survival probabilities. An univariable Cox proportional hazard model was used to model time to the composite outcome of lost to follow-up (LTFU) and death after TB diagnosis in the three subgroups with TB. Univariable and multivariable Cox proportional hazards models were fitted for patients with EPTB (period starting from the date of first diagnosis of EPTB) to explore associations of a priori defined variables (sex, age, CD4+ cell count at baseline, WHO stage at baseline, cART and CNS manifestations of TB) with the composite outcome of LTFU and death. The age variable was dichotomized at the upper quartile of the distribution of the cohort. The proportional hazard assumption was checked using Schoenfeld residuals. Missing data were excluded (complete case analysis). All statistical analyses were performed using Stata 12.1 (StataCorp, USA).

### Ethical considerations

All participants involved in this study gave written informed consent for the collection, storage and use of clinical data for research purposes within KIULARCO. The study protocol of KIU-LARCO has been approved by the Ifakara Health Institute Institutional Review Board, the National Institute of Medical Research in Tanzania, the Tanzanian Commission of Science and Technology as well as the Ethics committee of Northwest and Central Switzerland.

## Results

### Study population

In total, 3,620 patients were enrolled into KIULARCO between January 1st 2013 and December 31st 2017. Of these, we excluded 363 patients <18 years of age and 128 patients without any follow-up visits, resulting in 3,129 patients included in the final analysis (Fig 1). No characterization of the 128 patients excluded was done due to high rates of missing values. The median follow-up time was 1.25 years (IQR 0.46–2.85, range 3 days to 5 years) resulting in a total of 5370 person-years of follow-up.

Baseline characteristics of patients at the time of enrolment into KIULARCO according to TB group are shown in Table 1. The median age ranged from 36.5 years (IQR 31.3, 42.6) in the EPTB group to 39.9 years (IQR 34.0, 46.8) in the PTB group. The majority of patients (64.7%) were females, only the subgroup of patients with combined PTB and EPTB comprised more males (54.8%). The HIV infection at enrolment was more advanced in patients with a consecutive diagnosis of TB at any location: WHO stage 3/4 was attributed to 34.7% in the no TB

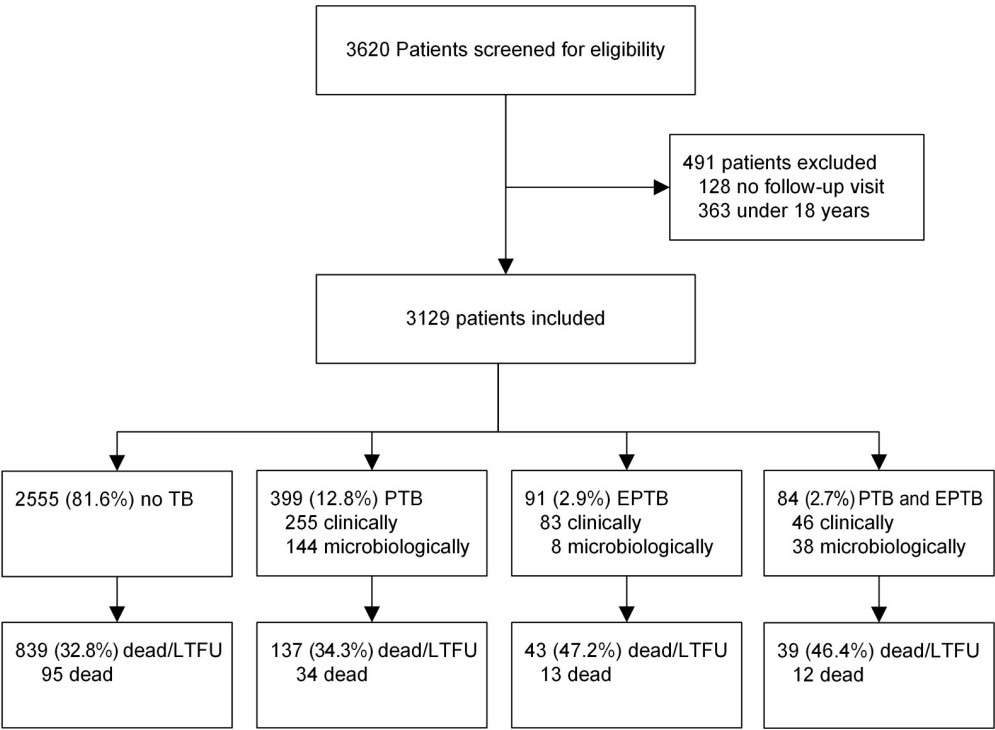

**Fig 1. Flowchart of patients included in the study.** Patients screened and included in the study. After inclusion, patients were allocated to 4 groups: patients with no tuberculosis (no TB), patients with pulmonary Tuberculosis (PTB), patients with extrapulmonary tuberculosis (EPTB) and patients with pulmonary and extrapulmonary tuberculosis. LTFU, lost to follow-up.

**Table 1. Baseline characteristics.**

| | EPTB | PTB | PTB and EPTB | no TB | p-value |
|---|---|---|---|---|---|
| **N (total = 3129)** | **91** | **399** | **84** | **2555** | |
| Microbiologically confirmed TB | 8 (8.8%) | 144 (36.1%) | 34 (40.4%)[£], 20 (23.8%)[$] | - | |
| Positive Xpert® MTB/RIF | 7 (7.7%) | 84 (21.0%) | 16 (19%)[£] / 12 (14.3%)[$] | - | |
| Year of registration | | | | | <0.001 |
| 2013 | 15 (16.5%) | 57 (14.3%) | 8 (9.5%) | 332 (13.0%) | |
| 2014 | 16 (17.6%) | 112 (28.1%) | 17 (20.2%) | 476 (18.6%) | |
| 2015 | 26 (28.6%) | 133 (33.3%) | 32 (38.1%) | 630 (24.7%) | |
| 2016 | 17 (18.7%) | 64 (16.0%) | 9 (10.7%) | 548 (21.4%) | |
| 2017 | 17 (18.7%) | 33 (8.3%) | 18 (21.4%) | 569 (22.3%) | |
| Age, years, median (IQR) | 36.5 (31.3, 42.6) | 39.9 (34.0, 46.8) | 39.9 (33.6, 45.5) | 38.0 (31.2, 46.2) | 0.007 |
| Sex, female | 55 (60.4%) | 214 (53.6%) | 38 (45.2%) | 1716 (67.2%) | <0.001 |
| Marital status | | | | | 0.001 |
| Married/living in relationship | 54 (59.3%) | 205 (51.4%) | 54 (64.3%) | 1581 (61.9%) | |
| Living alone | 37 (40.7%) | 194 (48.6%) | 30 (35.7%) | 974 (38.1%) | |
| Smoking, current | 8 (8.8%) | 37 (9.3%) | 7 (8.3%) | 102 (4.0%) | <0.001 |
| Missing information | 2 (2.2%) | 14 (3.5%) | 2 (2.4%) | 233 (9.1%) | |
| Alcohol consumption, current | 14 (15.4%) | 67 (16.8%) | 9 (10.7%) | 297 (11.6%) | 0.12 |
| Missing information | 2 (2.2%) | 14 (3.5%) | 2 (2.4%) | 233 (9.1%) | |
| HIV WHO stage | | | | | <0.001 |
| 1 | 14 (15.4%) | 45 (11.3%) | 9 (10.7%) | 1086 (42.5%) | |
| 2 | 7 (7.7%) | 29 (7.3%) | 2 (2.4%) | 385 (15.1%) | |
| 3 | 27 (29.7%) | 238 (59.6%) | 37 (44.0%) | 588 (23.0%) | |
| 4 | 42 (46.2%) | 80 (20.1%) | 35 (41.7%) | 299 (11.7%) | |
| Missing information | 1 (1.1%) | 7 (1.8%) | 1 (1.2%) | 197 (7.7%) | |
| CD4+ cell count (/μl) | | | | | <0.001 |
| <100 | 30 (33.0%) | 160 (40.1%) | 29 (34.5%) | 459 (18.0%) | |
| 100–199 | 21 (23.1%) | 67 (16.8%) | 20 (23.8%) | 387 (15.1%) | |
| 200–499 | 25 (27.5%) | 83 (20.8%) | 21 (25.0%) | 837 (32.8%) | |
| > = 500 | 7 (7.7%) | 33 (8.3%) | 2 (2.4%) | 439 (17.2%) | |
| Missing information | 8 (8.8%) | 56 (14.0%) | 12 (14.3%) | 433 (16.9%) | |
| CD4+ count (/μl), median (IQR) | 159 (60, 303) | 113 (49, 280) | 128 (54, 220) | 262 (117, 450) | <0.001 |
| Comorbidities | | | | | |
| Anemia | 72 (80.0%) | 292 (73.2%) | 72 (85.7%) | 1189 (46.6%) | <0.001 |
| Thrombocytopenia | 15 (16.7%) | 58 (14.5%) | 13 (15.5%) | 300 (11.8%) | 0.18 |
| Undernutrition | 21 (23.3%) | 151 (37.8%) | 24 (28.6%) | 359 (14.1%) | <0.001 |
| Obesity | 1 (1.1%) | 11 (2.8%) | 2 (2.4%)[$] | 136 (5.3%) | 0.030 |
| Arterial Hypertension | 10 (11.1%) | 40 (10.0%) | 10 (11.9%)[$] | 377 (14.8%) | 0.059 |
| Cryptococcosis | 3 (3.3%) | 8 (2.0%) | 1 (1.2%) | 56 (2.2%) | 0.80 |
| Viral hepatitis | 4 (4.4%) | 26 (6.5%) | 6 (7.1%) | 140 (5.5%) | 0.73 |
| Acute kidney failure | 5 (5.6%) | 9 (2.3%) | 6 (7.1%) | 33 (1.3%) | <0.001 |
| Chronic kidney failure | 0 (0.0%) | 2 (0.5%) | 2 (2.4%) | 21 (0.8%) | 0.28 |
| Malaria | 8 (8.9%) | 55 (13.8%) | 10 (11.9%) | 206 (8.1%) | 0.002 |
| Lower respiratory tract infection | 4 (4.4%) | 60 (15.0%) | 7 (8.3%) | 195 (7.6%) | <0.001 |

EPTB, extrapulmonary tuberculosis; IQR, interquartile range; PTB, pulmonary tuberculosis; TB, tuberculosis; WHO, world health organisation

[£] positive sputum sample;

[$] positive extrapulmonary sample

group versus 79.7%, 75.9% and 85.7% of the PTB, EPTB and PTB/EPTB group, respectively. The highest median CD4+ cell count was observed in the no TB group with 262 cells/μl, whereas in all other groups the median count was <200 cells/μl (113, 159 and 128 cells/μl in the PTB, EPTB and PTB/EPTB group, respectively). Patients with EPTB were more frequently married and less frequently smokers.

Regarding comorbidities within the first three months after enrolment, anemia was highly prevalent in all groups, but more so in patients with TB with the highest prevalence in the PTB/EPTB group with up to 85.7%. Undernutrition was also more prevalent in TB groups (no TB 14.1%, PTB 37.8%, EPTB 23.3%, PTB/EPTB 28.6%).

## Prevalence and incidence of EPTB, PTB and combined PTB/EPTB

Overall, 574 (18.4%) patients were diagnosed with confirmed or clinical TB at any localization (Fig 1). An extrapulmonary manifestation was observed in 175 (5.6%) patients, whereof 84 (48%) had concurrent pulmonary TB (PTB and EPTB group) and 91 (52%) no signs of pulmonary TB (EPTB group). Exclusive pulmonary manifestations were documented in 399 (12.8%) patients (PTB group).

The diagnosis of EPTB was made a median of 3.5 (IQR 1–21.5, range 0–1181) days and PTB at 2 (IQR 1–17, range 0–1296) days after enrolment. Most cases of TB were diagnosed during the first 90 days of observation and thus classified as prevalent at enrolment: this was the case in 334 (83.7%) in PTB, 78 (85.7%) in EPTB and 71 (84.5%) in PTB/EPTB. Tuberculosis diagnoses occurring 3 months or later after enrolment (incident cases) were diagnosed at an incidence density rate of 17.9/1000py (95% CI 14.2–22.6) for PTB and 5.8/1000py (95% CI 4.0–8.5) for EPTB corresponding to 5 cases of PTB, and 13 cases of EPTB and EPTB/PTB each.

Analyzing the time point of diagnosis of TB in relation to start of cART, in the majority of patients (406/508; 71.8%) TB was diagnosed before start of cART (Fig 2). No further information from the clinical notes concerning possible TB-associated immune reconstitution inflammatory syndrome (TB-IRIS) was available. In the subset of patients with initiation of cART ahead of a diagnosis of an incident case of TB, the median time from initiation of cART to the diagnosis of PTB and EPTB was 285 days (IQR 128–659) and 305 days (IQR 114–752, p = 0.78), respectively, possibly representing unmasking IRIS cases.

## Clinical presentation and microbiological testing

Affected organs in the 175 patients (multiple per patient possible) diagnosed with EPTB or PTB/EPTB were pleural space in 42 (24.0%), lymph node in 41 (23.4%), pericardium in 32 (18.3%), central nervous system in 21 (12%), urogenital tract in 18 (10.3%), peritoneum in 17 (9.7%), miliary in 13 (7.4%), bone in 2 (1.1%) and other in 21 (12.0%) of patients.

In total, 178/483 (36.9%) of patients with a pulmonary manifestation (i.e. the PTB and the combined PTB/EPTB group) and 28/175 (16.0%) of patients with an extrapulmonary manifestation were confirmed microbiologically (Table 1).

In total, we tested 1,241 samples (of 807 patients) with Xpert® MTB/RIF, whereof 500 (40.2%) were of extrapulmonary origin. Overall, 157/1241 (12.7%) samples analyzed were positive. Sputum samples were positive in 132/741 (17.8%) samples, extrapulmonary samples in 25/500 (5.0%) samples (Table 2). Over the study period, the number of performed Xpert® MTB/RIF tests per year increased for all materials with maximum usage in 2017 (S1 Dataset).

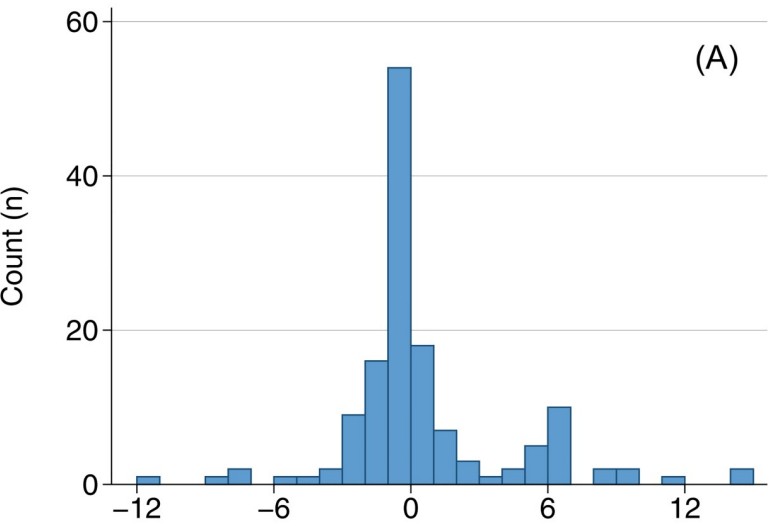

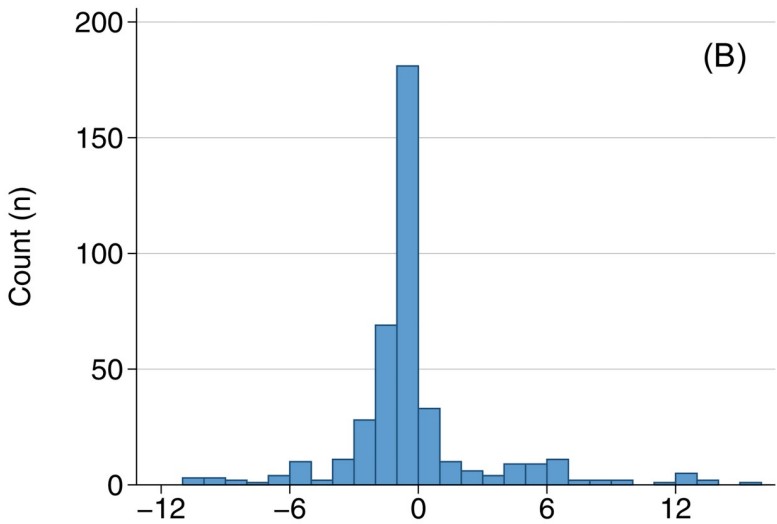

**Fig 2. Interval between start of antiretroviral treatment and diagnosis of tuberculosis.** (A) extrapulmonary tuberculosis (EPTB), (B) pulmonary tuberculosis (PTB). Time point "0" denotes the initiation of cART, negative values indicate that the diagnosis of TB was preceding the initiation of cART.

## Treatment and outcome

Combined antiretroviral therapy (cART) according to national guidelines was initiated in 2,786 (89.0%) of included patients after a median of 6 days (IQR 1–19 days, 90th percentile 102 days) from enrolment in the cohort. Out of the 574 patients with any diagnosis of TB, 508 (88.5%) started cART after a median of 23 (IQR 10–56) days. In the subset of patients with EPTB (including PTB/EPTB), 148 (82%) started cART after a median of 20 (IQR 7–40) days after enrolment. The most frequent combinations of antiretrovirals initiated were tenofovir disoproxil/lamivudine/efavirenz in 1,911 (68.5%) and tenofovir disoproxil/emtricitabine/efavirenz in 423 (15.2%) of patients started on cART.

**Table 2. Count and proportion of positive GeneXpert MTB/RIF by patient group (multiple tests per patients possible).**

|  | EPTB | PTB | PTB and EPTB | No TB | Total |
|---|---|---|---|---|---|
| Sputum | 0/28 | 106/275 (38.5%) | 26/42 (61.9%) | 0/396 | 132/741 (17.8%) |
| CSF | 2/10 (20%) | 0/12 | 0/9 | 0/72 | 2/103 (1.9%) |
| Pleural fluid | 0/6 | 0/9 | 5/18 (27.7%) | 0/0 | 5/33 (15.2%) |
| Ascitic fluid | 1/4 (25%) | 0/0 | 0/3 | 0/11 | 1/18 (5.6%) |
| Urine | 4/24 (16%) | 0/85 | 13/46 (28.3%) | 0/188 | 17/343 (5%) |
| Lymph node | 0/0 | 0/0 | 0/1 | 0/2 | 0/3 |

multiple tests per patient possible

CSF, cerebrospinal fluid; EPTB, extrapulmonary tuberculosis; PTB, pulmonary tuberculosis; TB, tuberculosis

Table 3 and Fig 3 show the numbers of patients who died or were lost to follow-up. Overall, 154 (4.9%) patients died (14.3% of patients with EPTB, 8.5% of patients with PTB and 3.7% of patients without TB). The combined endpoint of death/LTFU was most frequent in patients with EPTB (47.2% for EPTB alone, 46.4% for PTB/EPTB). The crude hazard ratios [HR (95% CI)] for death/LTFU after a diagnosis of TB were increased in the EPTB [HR 1.63, (1.14–2.31); p = 0.006] and the combined EPTB and PTB subgroup [HR 1.65, (1.15–2.46); p = 0.006] in comparison with the PTB subgroup (Fig 3).

Cox proportional hazard models were fitted using the data of all 175 with an EPTB (EPTB and PTB/EPTB group) to analyze factors associated with the composite outcome of death and LTFU (Table 4). In both the univariable and multivariable model, patients with EPTB, who have ever received cART had a significantly decreased risk of death/LTFU compared to patients who were not yet on cART (HR 0.15, 95% CI 0.08–0.27). The multivariable hazard ratios for death/LTFU were significantly increased for patients aged >45 years (HR 1.95, 95% CI 1.15–3.3).

## Discussion

In this rural sub-Saharan African cohort of PLHIV we found an EPTB prevalence of 5.6% with a high mortality of 14.3% (25/175) and a combined death/LTFU rate of 46.8%. Diagnosis remains a major challenge as only 36.9% of all pulmonary TB manifestations and 16% of extra-pulmonary manifestations were confirmed by microbiology. Patients with EPTB not receiving cART and with an age >45 years had a higher risk for a poor outcome.

The overall TB prevalence of 18.4% in our study is comparable to the UNAIDS Report 2018 showing a TB prevalence of 16% in PLHIV [28]. However, the 30% of EPTB within TB cases is considerably higher than rates documented worldwide and in Tanzania (14%, 22%, respectively) [1]. Studies from other rural sub-Saharan African settings, e.g. Ethiopia, also found a high EPTB prevalence in PLHIV (36.8%) [29]. Reasons might be the high number of patients presenting with advanced HIV (65%) and co-factors such as undernutrition [19, 30], which

**Table 3. Composite outcome of death/LTFU.**

|  | EPTB (n = 91) | PTB (n = 399) | EPTB and PTB (n = 84) | No TB (n = 2555) | p-value |
|---|---|---|---|---|---|
| Dead/LTFU | 43 (47.2%) | 137 (34.3%) | 39 (46.4%) | 839 (32.8%) | 0.003 |
| Dead | 13 (14.3%) | 34 (8.5%) | 12 (14.3%) | 95 (3.7%) | <0.001 |

EPTB, extrapulmonary tuberculosis; LTFU, lost to follow-up; PTB, pulmonary tuberculosis; TB, tuberculosis

p-values indicated are derived by multiple group comparison using chi-square tests.

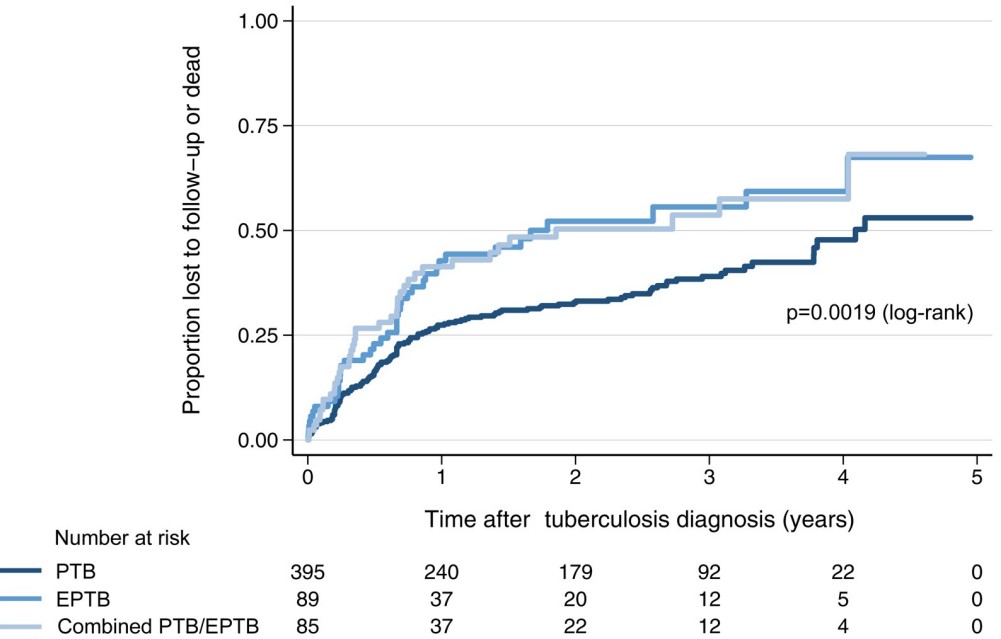

| Number at risk | | | | | | |
|---|---|---|---|---|---|---|
| PTB | 395 | 240 | 179 | 92 | 22 | 0 |
| EPTB | 89 | 37 | 20 | 12 | 5 | 0 |
| Combined PTB/EPTB | 85 | 37 | 22 | 12 | 4 | 0 |

**Fig 3. Kaplan-Meier curves for composite outcome of lost to follow-up/dead in patients with tuberculosis.**

has been demonstrated to be 30% in the same cohort [31] and is more prevalent in rural than in urban settings [32]. Over the study duration, the proportion of patients with a diagnosis of EPTB in all patients with TB increased from 28.8% in 2013 to 51% in 2017 patients. This might be likely due to reporting bias, as we concurrently did a prospective study on the impact of sonography for EPTB diagnosis, leading to a more sensitive evaluation of patients with TB

**Table 4. Cox proportional hazard model for composite outcome of LTFU/death in EPTB patients (with or without pulmonary manifestation).**

| | Univariable | | Multivariable | |
|---|---|---|---|---|
| | **HR (95% CI)** | **p** | **HR (95% CI)** | **p** |
| Female sex [a] | 0.86 (0.52, 1.41) | 0.541 | 0.86 (0.52, 1.41) | 0.541 |
| CD4+ cell count at baseline [cells/ul] | | | | |
| <200 | *Reference* | | *Reference* | |
| ≥200 | 0.87 (0.53, 1.42) | 0.59 | 0.94 (0.56, 1.58) | 0.829 |
| WHO stage at baseline | | | | |
| I & II | *Reference* | | *Reference* | |
| III & IV | 1.10 (0.62, 1.97) | 0.743 | 0.88 (0.46, 1.67) | 0.693 |
| Tuberculous meningitis [b] | 0.91 (0.44, 1.89) | 0.805 | 0.71 (0.33, 1.50) | 0.363 |
| ART intake [c] | 0.19 (0.11, 0.32) | <0.001 | 0.15 (0.08, 0.27) | <0.001 |
| Age > 45 years [d] | 1.35 (0.84, 2.16) | 0.214 | 1.95 (1.15, 3.30) | 0.013 |

ART, antiretroviral therapy; CI, confidence intervall; HR, hazard ratio; LTFU, lost to follow-up; WHO world health organization

Reference groups:

[a] male sex,

[b] no tuberculous meningitis,

[c] no ART intake,

[d] age ≤ 45 years

[25]. In other settings, implementation of cART has led to a decrease in EPTB rates in PLHIV [33].

The mortality of 14.3% of EPTB patients is lower than a recent study in urban setting in Ghana with a mortality of 28.7% [34]. However, the rate of 14.3% we found is comparable to a multicohort study comparing clinical outcomes of EPTB in 22 ART programs with a mortality of 11.4% in PLHIV with EPTB [35]. Comparison between different studies remains a challenge, as diagnosis of EPTB is not very standardized and mostly done on clinical parameters only [25].

In our study, the rate of positive Xpert® MTB/RIF was low with 12.7% (157/1,241 samples analyzed). While the majority of positive samples (86.3%) originated from sputum, the rate of positive samples from extrapulmonary sites was very low with 5% (25/500). Poor sensitivity of Xpert® MTB/RIF has been demonstrated in PLHIV with HIV previously [5] and is partly due to challenges in proper sampling [36] and the paucibacillary nature of disease in PLHIV and in extrapulmonary sites [37, 38]. In one study overall sensitivity and specificity of Xpert® MTB/ RIF compared to culture was 81.3% and 99.8%, respectively in patients with EPTB. However, when compared to a combination of culture and clinical diagnosis, the sensitivity of Xpert® MTB/RIF in cavitary fluids was under 50% [39]. The inclusion of the clinical diagnosis into a combined gold standard was based on post-mortem studies in PLHIV in resource-limited settings, which have shown a prevalence of TB in sub-Saharan Africa up to 43.2% [40]. A recent meta-analysis found a sensitivity of Xpert® MTB/RIF in pleural fluid of 50.9% and documents the important role of Xpert® MTB/RIF as confirmatory test in extrapulmonary samples, while exclusion remains a challenge [8].

While the majority of TB cases (71.8%) were diagnosed before the start of cART, a minority (9.1%) were diagnosed in first 30 days after, possibly presenting an unmasking IRIS. This hyper-responsiveness due to an increase in T-cells [41] has been reported to occur in 36–54% of TB/HIV co-infected patients [42, 43] and usually occurs within the first month after start of the antiretroviral therapy [44]. Patients presenting late after cART start with incident TB were few–unfortunately we could not analyze details on cART adherence or virological suppression as routine viral load was not implemented during the study period.

In a meta-analysis of PLHIV in sub-Saharan Africa around 40% of patients were LTFU five years after initiation of cART with death rates approaching 15% [45]. In our cohort this number was even higher in those with EPTB diagnosis. A recent meta-analysis including data of 7,377 patients who started cART and were subsequently LTFU showed a rate of 21.8% of subjects to be deceased while 14.8% of patients LTFU have been transferred to another clinic [46]. Factors associated with poor outcome in patients with EPTB in our study were age of >45 years and not being on cART. Both confirm previous studies, where age has been shown as a risk factor for disseminated TB and death [47, 48], while patients with EPTB co-infected with HIV without cART showed poorer outcomes than with cART [4].

The adjusted model did not show an association between indicators of HIV associated immunosuppression (WHO stage and CD4+ cell count) at baseline and the combined endpoint. This observation might be due to the inclusion of only patients developing EPTB into the model. By analyzing a selected group with a high probability of the endpoint as well as a presumed high level of functional immunodeficiency, WHO stage and CD4+ cell count might not further discriminate risk.

A major strength of this study is the prospective assessment of TB by means of ICD-10 codes in a long-term cohort of PLHIV ensuring standardized reporting. Furthermore, this study depicts a sub-Saharan African rural setting representative of settings with limited diagnostic resources available to clinicians and a high proportion of diagnoses relying on clinical presentation only. Those conditions are neglected as many studies have been done in urban settings.

The clinical diagnosis of EPTB determined by the physician, however remains a major limitation, as it is a poorly defined entity which might have been assessed differently by physicians. Patients with important differential diagnoses such as cancer, heart failure, lung abscess or other diseases might have been missed. As in other studies from African settings [4, 5, 49, 50] the most frequent sites of EPTB were pleurisy and lymphadenopathy. As the lymphadenopathy was a clinical diagnosis without histological confirmation, the assumption of TB as underlying disease might have been overestimated.

Besides improving microbiological detection of TB, e.g. by Xpert MTB/RIF Ultra [51], lateral flow urine lipoarabinomannan assay (LF-LAM) [52] or adenosine deaminase [53, 54], the clinical evaluation might benefit from integration of an additional standardized testing, such as e.g. Focused Assessment with Sonography for HIV and Tuberculosis (FASH) [25, 55].

Another limitation is the rather low observation period of patients after start on cART (median 1.25 years), mainly due to the high number of patients LTFU, which precluded an exact estimation of the true mortality rate.

In conclusion, EPTB remains a relevant comorbidity with a high mortality and LTFU in a rural sub-Sahara African setting. Establishing a microbiological diagnosis of EPTB in the absence of culture and histopathology remains a major challenge in settings, where invasive diagnostic procedures are not routinely performed. Despite the increasing access to Xpert® MTB/RIF MTB/RIF in sub-Saharan Africa it remains unclear to what extent this translates in clinical benefit [56]. The lack of ART as a factor for poor prognosis points at the importance of early ART start. Further research needs to be conducted to better define clinical predictors and new microbiological tests to improve early detection and outcome of EPTB.

## Supporting information

**S1 Dataset.**
(XLSX)

**S1 Table. Definitions of tuberculosis.**
(DOCX)

## Acknowledgments

We thank all patients who participated in KIULARCO and all staff members of the Chronic Disease Clinic at the St. Francis Referral Hospital, Ifakara, Tanzania. We thank all the members of the KIULARCO study group, who are: Aschola Asantiel, Farida Bani, Manuel Battegay, Theonestina Byakuzana, Adolphina Chale, Anna Eichenberger, Gideon Francis, Hansjakob Furrer, Anna Gamell, Tracy Glass, Speciosa Hwaya, Aneth V Kalinjuma, Joshua Kapunga, Bryson Kasuga, Andrew Katende, Namvua Kimera, Yassin Kisunga, Thomas Klimkait, Ezekiel Luoga, Herry Mapesi, Slyakus Mlembe, Mengi Mkulila, Margareth Mkusa, Dorcas K Mnzava, Getrud J Mollel, Lilian Moshi, Germana Mossad, Dolores Mpundunga, Athumani Mtandanguo, Selerine Myeya, Sanula Nahota, Robert C Ndege, Omary Rajabu Ngome, Agatha Ngulukila, Alex John Ntamatungiro, Amina Nyuri, Daniel H Paris, Leila Samson, Elizabeth Senkoro, Jenifa Tarimo, Yvan Temba, Juerg Utzinger, Fiona Vanobberghen, Maja Weisser, John Wigay, Herieth Wilson.

We thank the Government of the Canton of Basel, Switzerland, the Swiss Tropical & Public Health Institute, the Ifakara Health Institute, the University Hospital Basel, the Government of Tanzania, and the United States Agency for International Development through TUNAJALI-Deloitte/USAID Boresha Afya for support of the CDCI. Furthermore, we thank the "Freiwillige Akademische Gesellschaft Basel" for the financial support.

## Author Contributions

**Data curation:** Fabian Christoph Franzeck, Tracy Renee Glass, Farida Bani, Omary Ngome Rajab.

**Formal analysis:** Fabian Christoph Franzeck.

**Investigation:** Armon Arpagaus, Farida Bani, Omary Ngome Rajab.

**Methodology:** Armon Arpagaus, Omary Ngome Rajab, Maja Weisser.

**Supervision:** Maja Weisser.

**Writing – original draft:** Armon Arpagaus, Fabian Christoph Franzeck, Maja Weisser.

**Writing – review & editing:** Armon Arpagaus, Fabian Christoph Franzeck, George Sikalengo, Robert Ndege, Dorcas Mnzava, Martin Rohacek, Jerry Hella, Klaus Reither, Manuel Battegay, Tracy Renee Glass, Daniel Henry Paris, Omary Ngome Rajab, Maja Weisser.

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
