## [Decision Letter · Decision Letter 0]

11 Nov 2019

PONE-D-19-26934

Extrapulmonary tuberculosis in HIV-infected patients in rural Tanzania. The prospective Kilombero and Ulanga antiretroviral cohort.

PLOS ONE

Dear PD Dr Weisser,

Thank you for submitting your manuscript to PLOS ONE. After careful consideration, we feel that it has merit but does not fully meet PLOS ONE’s publication criteria as it currently stands. Therefore, we invite you to submit a revised version of the manuscript that addresses the points raised during the review process.

We would appreciate receiving your revised manuscript by Dec 26 2019 11:59PM. To enhance the reproducibility of your results, we recommend that if applicable you deposit your laboratory protocols in protocols.io, where a protocol can be assigned its own identifier (DOI) such that it can be cited independently in the future. For instructions see: http://journals.plos.org/plosone/s/submission-guidelines#loc-laboratory-protocols

We look forward to receiving your revised manuscript.

Kind regards,

Marcel Yotebieng, M.D., MPH, Ph.D

Academic Editor

PLOS ONE

Journal Requirements:

2. One of the noted authors is a group or consortium: KIULARCO Study Group

In addition to naming the author group, please list the individual authors and affiliations within this group in the acknowledgments section of your manuscript. Please also indicate clearly a lead author for this group along with a contact email address.

Additional Editor Comments:

We hear back from two reviewers and both agree on the merit of your paper and to the fact that it requires some revision before it can be accepted. Please use their suggestions to revise and improve the readability of the paper. Also note that PLOS to not edit the paper before publication, so make sure the paper is proof read by a native English speaker.

Reviewers' comments:

Reviewer's Responses to Questions

**Comments to the Author**

1. Is the manuscript technically sound, and do the data support the conclusions?

Reviewer #1: Yes

Reviewer #2: Yes

2. Has the statistical analysis been performed appropriately and rigorously? 

Reviewer #1: I Don't Know

Reviewer #2: No

3. Have the authors made all data underlying the findings in their manuscript fully available?

Reviewer #1: Yes

Reviewer #2: Yes

4. Is the manuscript presented in an intelligible fashion and written in standard English?

Reviewer #1: Yes

Reviewer #2: Yes

5. Review Comments to the Author

Reviewer #1: REVIEW OF MANUSCRIPT OCTOBER 2019

Extrapulmonary tuberculosis in HIV-infected patients in rural Tanzania. The

prospective Kilombero and Ulanga antiretroviral cohort.

GENERAL REMARKS

This prospective study examines extrapulmonary TB (EPTB) in HIV positive patients in Tanzania. The paper is well written and has a number of strengths. It will benefit from proof-reading to correct a few sentences where the words do not read well. Some suggestions are highlighted below and in the paper for the authors’ consideration to improve the paper.

TITLE

Instead of a full stop after Tanzania, a colon would be more appropriate as follows: Extrapulmonary tuberculosis in HIV-infected patients in rural Tanzania: The prospective Kilombero and Ulanga antiretroviral cohort

ABSTRACT

Reading through the paper, it appears that in addition to characterizing the epidemiology and clinical characteristics of these patients, the key outcomes of interest are death / loss to follow up rather than the broad term clinical outcomes which would also include outcomes such as cure. The authors should consider rewording the aim/objective to reflect the interest in the predictors of the outcome lost to follow up / death.

INTRODUCTION

The introduction is generally well written.

Page 4 Lines 80-81 My comment on the aim of the study highlighted under the ABSTRACT portion above also applies. The authors should consider rewording the aim/objective to reflect that they examined predictors of lost to follow up / death.

METHODS

Please indicate at the beginning of the Methods section that this is a prospective of study so that readers are aware of this methodology right from the start even before providing information on the study site and patient population.

It would be helpful to readers to have a good understanding of the study setting since the authors highlight that it is rural Tanzania. So even though the authors provide references for details on the cohort, providing a little bit of information on the setting and the context of these patients would be very useful for readers especially later on when the results show such high rates of lost to follow up.

Page 4 89-90 Please highlight what the guideline was for starting cART for HIV patients in the program.

Page 5 Lines 114-115 I found the following phrase “1 week before until 3 months after working diagnosis” challenging to understand. Is it to say that patients were designated as PTB even one week before they were assigned a diagnosis of TB? Please explain the phrase so that it is self-explanatory.

Page 6 Lines 151-152 Please explain how the incidence density was calculated

RESULTS

Page 7 Line 175 Please provide the range of the follow up period as well.

Page 7 Lines 179 – 188. The authors make statements such as “more advanced” and “no major differences”. It is not clear whether these inferences are being made from just eyeballing the figures in Table 1. Please indicate whether these differences were statistically significant or not statistically significant.

In Table 1, I noticed that registration was highest in 2015 and I am wondering whether there was some reason for this. The authors should please point out this finding and comment on it in the discussion.

Page 10 Line 209 For the incident TB cases, please indicate how many patients were diagnosed with TB 90 days after enrolment and the range of the period between the time of enrollment and the diagnosis of TB.

Please provide information on the prevalence of CNS manifestation in the EPTB patients.

Please indicate what the reference groups are for all the variables in Table 4

DISCUSSION

The authors indicate a high mortality of 14.3% among their EPTB patients. How does this finding compare with what other studies have found?

Given the various variables that the authors report on in their study, there is opportunity for rich discussion on the study findings. The authors should include more discussion on the poor outcome of their patients LTFU in view of the clinical characteristics reported on such as WHO stage, comorbidities and CD4 count.

In the conclusion, the authors highlight that lack of ART and age >45 years were risk factors for death or loss to follow-up. What recommendations do the authors offer for clinicians or program managers to address these findings?

Reviewer #2: Review report

Title : “Extrapulmonary tuberculosis in HIV-infected patients in rural Tanzania. The prospective Kilombero and Ulanga antiretroviral cohort.”

Title

1) I don’t understand why you have put “full stop” after Tanzania. Why not “:” ?

Abstract

1) Please you should explain all abbreviations at the first appearance in the text e.g: AFB, py, CI

2) You stated that: “The combined endpoint of death and LTFU was observed in 1058 (33.8%) patients, most frequently in the subgroup of EPTB (47.2%)”.

You should give hazard ratio

3) “(HR 1.95, 95%CI 1.15-3.3; p = 0.013)”: This is not well written.

You can for example state after hazard ratio [HR (95% CI)] and then after each risk factor you just need to put the values.

Introduction

1) Line 55-56: “…14% incident cases” where?

2) Line 71: Heart failure is not the classical differential diagnosis of pleural TB. Heart failure gives typically a transudate pleural effusion whereas pleural TB gives exudative pleural effusion. You should remove heart failure as a differential diagnosis.

Methods

1) Line 101: Which sites (localizations) did you use for sonography screening? You should specify.

2) The Methods section should begin by specify the group of patients included. You have put it at the end of the first paragraph.

3) I don’t understand the rationale of “antituberculous drugs started” one week before the working diagnosis. This claim is confusing me.

4) The section named definitions spanned data collection. You should rename or separate data collection and operational definitions sections.

5) All the definitions used are very hard to understand and are beat confusing. Can you specify the definitions used to create all the ICD10 codes?

6) The major issue in this chapter is the absence of histopathological features.

7) In HIV environment we usually used 90 days for LTFU definition and not 60 days. What are the rationales of 60 days choice?

8) Which statistical test did you use to compare Kaplan-Meier curves (Log-rank test?)

Results

1) What are the characteristics of 128 excluded patients in comparison of the characteristics of included ones?

2) Did you verify that the patients classified as LTFU were not died?

3) You have compared baseline characteristics in different groups without given p-value for comparison. The p-value is not specified in the Table 1.

4) The first column in your Table 1 should be EPTB and you should compare the other forms to this one.

5) Table 1 is not well formatted which horizontal lines inside the table.

6) Line 199: You should begin by EPTB

7) Is it any difference between the median time from initiation of cART to the diagnosis of PTB and EPTB

8) Line 222: I hope that the localization of EPTB was specified in all of your participants.

9) Lines 226 and 227: pulmonary and extrapulmonary manifestations refer to PTB and EPTB? Not clear.

10) Table 2: please correct “Lymph node”

11) Lines 240 and 241: please rephrase sentence. The second part is not understandable

12) Lines 250-251: EPTB and EPTB/PTB were compare to what (PTB or no TB)?

13) Figure 2 is difficult to understand. Negative month means what?

14) I did not see the verification of proportional hazard models assumptions.

Discussion

The discussion section is very well written.

6. PLOS authors have the option to publish the peer review history of their article (what does this mean?). If published, this will include your full peer review and any attached files.

Reviewer #1: Yes: Sally-Ann Ohene

Reviewer #2: Yes: Eric Walter PEFURA YONE

---

## [Author Response · Author response to Decision Letter 0]

17 Dec 2019

Please find the point to point reply in the Response to Reviewers letter as well.

TITLE

Reviewer #1

Instead of a full stop after Tanzania, a colon would be more appropriate as follows: Extrapulmonary tuberculosis in HIV-infected patients in rural Tanzania: The prospective Kilombero and Ulanga antiretroviral cohort.

Reviewer #2

1) I don’t understand why you have put “full stop” after Tanzania. Why not “:” ?

 ANSWER: We agree with this consideration and have changed the title accordingly.

ABSTRACT

Reviewer #1

Reading through the paper, it appears that in addition to characterizing the epidemiology and clinical characteristics of these patients, the key outcomes of interest are death / loss to follow up rather than the broad term clinical outcomes which would also include outcomes such as cure. The authors should consider rewording the aim/objective to reflect the interest in the predictors of the outcome lost to follow up / death.

 ANSWER: We reworded to ‘risk factors of poor outcome’.

Reviewer #2

1) Please you should explain all abbreviations at the first appearance in the text e.g: AFB, py, CI

 ANSWER: We thank for this suggestion and have corrected the abbreviations.

2) You stated that: “The combined endpoint of death and LTFU was observed in 1058 (33.8%) patients, most frequently in the subgroup of EPTB (47.2%)”. You should give hazard ratio.

ANSWER: We constructed a cox model exclusively for patients with EPTB, so we could investigate risk factors for the combined endpoint in these patients. Therefore, no hazard ratios for comparisons of different study groups (e.g. EPTB vs. PTB) can be indicated, as the cox model did never include patients of e.g. the PTB group. However, with the data given in Table 3, the frequencies of the combined endpoint in the different study groups can be compared. We added details on the test used for the calculation of p-values in the table legend, in addition to the preexisting mention in the methods section.

3) “(HR 1.95, 95%CI 1.15-3.3; p = 0.013)”: This is not well written.

You can for example state after hazard ratio [HR (95% CI)] and then after each risk factor you just need to put the values.

 ANSWER: We agree with this proposal and adjusted the writing.

INTRODUCTION

Reviewer #1

Page 4 Lines 80-81 My comment on the aim of the study highlighted under the ABSTRACT portion above also applies. The authors should consider rewording the aim/objective to reflect that they examined predictors of lost to follow up / death.

ANSWER: We corrected as suggested (page 4, line 80).

Reviewer #2

1) Line 55-56: “…14% incident cases” where?

ANSWER: According to the Tuberculosis Report 2018 it applies to the incidence globally. We specified accordingly.

2) Line 71: Heart failure is not the classical differential diagnosis of pleural TB. Heart failure gives typically a transudate pleural effusion whereas pleural TB gives exudative pleural effusion. You should remove heart failure as a differential diagnosis.

ANSWER: We thank for this feedback. In remote rural settings, the laboratory mostly cannot provide a detailed analysis of fluid allowing differentiation between a transudate and an exudative character. Therefore, heart failure remains a possible differential diagnosis of pleural effusion. We have added the word ‘clinical’ to stress the constraint of diagnostic options.

METHODS

Reviewer #1

Please indicate at the beginning of the Methods section that this is a prospective of study so that readers are aware of this methodology right from the start even before providing information on the study site and patient population. It would be helpful to readers to have a good understanding of the study setting since the authors highlight that it is rural Tanzania. So even though the authors provide references for details on the cohort, providing a little bit of information on the setting and the context of these patients would be very useful for readers especially later on when the results show such high rates of lost to follow up.

ANSWER: We thank the reviewer for this comment and added more details in the highlighted manuscript on page 4 lines 85ff.

Page 4 89-90 Please highlight what the guideline was for starting cART for HIV patients in the program.

ANSWER: We added references for the guidelines used, which are the current “national guidelines for the management of HIV and AIDS” (NACP), issued by the ministry of health of Tanzania.

During the study period the WHO guidelines recommendations and its derived NACP guidelines in start of antiretroviral drugs changed from HIV WHO stage III or IV or CD4 cell counts <350cells/mm3, to <500cells/mm3 to treatment of all HIV-positive individuals (page 4, line 93ff).

Page 5 Lines 114-115 I found the following phrase “1 week before until 3 months after working diagnosis” challenging to understand. Is it to say that patients were designated as PTB even one week before they were assigned a diagnosis of TB? Please explain the phrase so that it is self-explanatory.

ANSWER: We thank for this question. This sentence illustrates the difficulty that some patients started the treatment on a “probatory” base as the diagnosis of tuberculosis is often clinical only. Thus, the treatment might have started before the indication of an ICD-10 Code. We added working diagnosis of ‘suspected TB’ to make this clearer on page 5, line 120.

Page 6 Lines 151-152 Please explain how the incidence density was calculated.

ANSWER: The incidence density was calculated as (number of incident cases)/(person-time of subjects at risk in the unit of person-years). This contrasts with incidence rates (number of incident cases/numbers of subjects observed), which don’t incorporate a time dimension into the denominator. We added this in the definitions section on page 6, line 164.

Reviewer #2

1) Line 101: Which sites (localizations) did you use for sonography screening? You should specify.

ANSWER: We performed the sonography screening according to the Focused Assessment with Sonography for HIV and Tuberculosis protocol (FASH) and added the reference to the manuscript. The pathological findings screened for in the FASH-protocol were pleural or pericardial effusion, ascites, abdominal lymph nodes, hypoechogenic lesions in liver/spleen, Ileum wall thickening or ileum wall destruction (added on page 5, lines 109-111).

2) The Methods section should begin by specify the group of patients included. You have put it at the end of the first paragraph.

ANSWER: We thank for this correction, which was implemented in the manuscript.

3) I don’t understand the rationale of “antituberculous drugs started” one week before the working diagnosis. This claim is confusing me.

ANSWER: We thank the reviewer for this question. Please see our answer above to the same point raised by reviewer #1. Some patients started the treatment on clinical suspicion only, while microbiological prove follows later. We have added the explanation due to the possible premature start of TB-treatment before an indication of an ICD-10 diagnosis

4) The section named definitions spanned data collection. You should rename or separate data collection and operational definitions sections.

ANSWER: We reviewed the section entitled ‘definitions’ and believe the points mentioned should remain here as they are rather part of the analytical process than the prospective data collection.

5) All the definitions used are very hard to understand and are a bit confusing. Can you specify the definitions used to create all the ICD10 codes?

ANSWER: We agree it is a long paragraph, but we believe these definitions are key specifically for the field of the clinically suspected cases, which in daily life are done at the discretion of the treating physician. As we mentioned in the discussion this remains a limitation of the study. To simplify, we shortened the paragraph on definition of comorbidities on page 6, line 141 ff. 

6) The major issue in this chapter is the absence of histopathological features.

ANSWER: We agree and have pointed this out in the methods page 5, line 116/117. Additionally, this is mentioned in the discussion as a limitation (page 14, line 336ff).

7) In HIV environment we usually used 90 days for LTFU definition and not 60 days. What are the rationales of 60 days choice?

ANSWER: When the definition for LTFU for use in this cohort was designed, we incorporated empirical evidence about the choice of the optimal cutoff to predict a true LTFU status. This study can be accessed at https://www.ncbi.nlm.nih.gov/pubmed/20219765 . 

8) Which statistical test did you use to compare Kaplan-Meier curves (Log-rank test?)

ANSWER: We added a p-value derived by log-rank testing to Figure 3, comparing the 3 groups of patients with a diagnosis tuberculosis, including a mention of the test in the methods section (page 6, line 163).

The p-values already indicated in the text comparing risk factors for the composite outcome in patients with EPTB are derived from the survival analysis using the Cox proportional hazard model. We think this was appropriately indicated by mentioning p-values in exclusively in parenthesis after the respective hazard ratios. 

RESULTS

 COMMENT: Please note the change in the inclusion timeline on page 7, line 178; we replaced September 2013 with January 2013, which was a mistake in the first edition.

Reviewer #1

Page 7 Line 175 Please provide the range of the follow up period as well. 

ANSWER: Please find the range in the highlighted manuscript (3 days-5 years) on page 7, line 182.

Page 7 Lines 179 – 188. The authors make statements such as “more advanced” and “no major differences”. It is not clear whether these inferences are being made from just eyeballing the figures Table 1. Please indicate whether these differences were statistically significant or not statistically significant.

ANSWER: We thank for this comment and added the p-values of comparison tests (according to the methods section) to table 1.

In Table 1, I noticed that registration was highest in 2015 and I am wondering whether there was some reason for this. The authors should please point out this finding and comment on it in the discussion.

ANSWER: We thank you for the observation and discussed the finding with all investigators. However, we could not find a convincing conclusion to explain this peak. Therefore, we believe, this might reflect natural fluctuation or factors not explainable with the data on hand. We have already commented this in the discussion on page 13, line 289ff.

 

Page 10 Line 209 For the incident TB cases, please indicate how many patients were diagnosed with TB 90 days after enrolment and the range of the period between the time of enrollment and the diagnosis of TB.

ANSWER: Out of a total of 574 patients with TB (399 PTB, 84 PTB/EPTB and 91 EPTB), 334 PTB, 71 PTB/ETPB and 78 EPTB cases were diagnosed within the 3 months (baseline prevalence). Thus, 5 cases of PTB, 13 EPTB and 13 cases of EPTB/PTB were diagnosed after these first 3 months. We added this information on page 10, line 215-216.)

The range of days has been added in the manuscript on page 10, line 210 and 211.

Please provide information on the prevalence of CNS manifestation in the EPTB patients.

ANSWER: The prevalence rates of varying organ manifestations of EPTB were described in the initial manuscript under subheading “Clinical presentation and microbiological testing” on page 10, line 227. This includes CNS manifestations as well, which were present in 12% of all cases of EPTB.

Please indicate what the reference groups are for all the variables in Table 4

ANSWER: Reference groups were added as footnotes to the table as requested.

Reviewer #2 

1) What are the characteristics of 128 excluded patients in comparison of the characteristics of included ones?

ANSWER: Those patients had high rates of missing values for most variables used in table 1, thus no clear description of these patients was possible. We added a comment in the manuscript on page 7, line 180.

2) Did you verify that the patients classified as LTFU were not died?

ANSWER: As LTFU commonly indicates a category of subjects, whose outcome is unknown, we cannot know if patients died. Indeed, a high death rate is described in studies actively tracking patients lost to follow-up (e.g. Egger et al PLoS Med. 2011 Jan 18;8(1):e1000390): Correcting mortality for loss to follow-up: a nomogram applied to antiretroviral treatment programs in sub-Saharan Africa.) In our setting we do regular tracking, but still remain with a high percentage of patients LTFU.

3) You have compared baseline characteristics in different groups without given p-value for comparison. The p-value is not specified in the Table 1.

ANSWER: The p-values have been added to table 1.

4) The first column in your Table 1 should be EPTB and you should compare the other forms to this one.

ANSWER: We have changed the order as requested.

5) Table 1 is not well formatted which horizontal lines inside the table.

ANSWER: The table was indeed already designed to be legible without the necessity of horizontal lines. We removed the horizontal lines in the updated version. 

6) Line 199: You should begin by EPTB

ANSWER: We thank for this correction and adjusted the title.

7) Is it any difference between the median time from initiation of cART to the diagnosis of PTB and EPTB

ANSWER: We added the p-value derived by a Mann–Whitney U test comparing the intervals to the results section, it was nonsignificant with a value of 0.78 (page 10, line 223).

8) Line 222: I hope that the localization of EPTB was specified in all of your participants.

ANSWER: In the 175 patients with an extrapulmonary manifestation the localization was always specified. The distribution of organ manifestations has already been described in the results section under “Clinical presentation and microbiological testing”. 

9) Lines 226 and 227: pulmonary and extrapulmonary manifestations refer to PTB and EPTB? Not clear.

ANSWER: We used “manifestation” instead of “PTB and EPTB” to account for the fact that patients in both the EPTB and the combined PTB/EPTB study group show signs of extrapulmonary TB. The same is true for pulmonary manifestations which are present in the PTB and the combined PTB/EPTB group. The respective denominator in the numbers indicated in the same sentence is the appropriate sum of the two groups. We further clarified this in the methods (page 6, line 139).

10) Table 2: please correct “Lymph node”

ANSWER: We thank for the comment and corrected accordingly.

11) Lines 240 and 241: please rephrase sentence. The second part is not understandable

ANSWER: We agree with the comment and clarified the sentence (now page 11, line 248-249)

12) Lines 250-251: EPTB and EPTB/PTB were compare to what (PTB or no TB)?

ANSWER: Table 3 describes a crude comparison of outcomes in the four study groups. The indication of one p-value per row indicates that it describes a multiple group comparison across all 4 groups and not the result of an individual between-two-group comparison. We added a specification of this in the table legend (page 12, line 263).

13) Figure 2 is difficult to understand. Negative month means what?

ANSWER: The interval denotes the difference between date of diagnosis of TB and the initiation of cART in months. Positive numbers indicate, that the diagnosis of TB was after the initiation of cART, negative numbers indicate, that the diagnosis of TB was before the initiation of cART. We added a sentence in the figure legend (‘Out of the 574 patients with any diagnosis of TB, 508 (88.5%) started cART after a median of 23 (IQR 10-56) days.’)

14) I did not see the verification of proportional hazard models assumptions.

ANSWER: In the initial manuscript (“Data management and analysis”, page 6, line 165ff) we indicated, that Schoenfeld residuals were used to verify the proportional hazards assumption.

We are not sure, whether the reviewer was asking to actually see the results of the assumption testing done. In our opinion, it is good standard in medical literature to explicitly indicate that assumption testing was performed and to mention the respective method used. However, to our knowledge, the results of such testing are not commonly published with the article. If of interest for the reviewer, we show the results of the formal statistical testing here.

Variable rho chi2 df Prob>chi2

Sex 0.08087 0.51 1 0.4755

CD4+ cell count at baseline 0.13093 1.33 1 0.2485

Tuberculous meningitis -0.18395 2.67 1 0.1022

ART intake -0.17803 3.14 1 0.0762

Age > 45 years 0.09997 0.83 1 0.362

global test 7.79 5 0.1683

ACKNOWLEDGEMENTS

COMMENT: Page 16, line 356. We added the members of the KIULARCO study group in the acknowledgements. The corresponding author is Prof. Dr. Maja Weisser.

---

## [Decision Letter · Decision Letter 1]

16 Jan 2020

PONE-D-19-26934R1

Extrapulmonary tuberculosis in HIV-infected patients in rural Tanzania: The prospective Kilombero and Ulanga antiretroviral cohort.

PLOS ONE

Dear PD Dr Weisser,

Thank you for submitting your manuscript to PLOS ONE. After careful consideration, we feel that it has merit but does not fully meet PLOS ONE’s publication criteria as it currently stands. Therefore, we invite you to submit a revised version of the manuscript that addresses the points raised during the review process.

We would appreciate receiving your revised manuscript by Mar 01 2020 11:59PM. To enhance the reproducibility of your results, we recommend that if applicable you deposit your laboratory protocols in protocols.io, where a protocol can be assigned its own identifier (DOI) such that it can be cited independently in the future. For instructions see: http://journals.plos.org/plosone/s/submission-guidelines#loc-laboratory-protocols

We look forward to receiving your revised manuscript.

Kind regards,

Marcel Yotebieng, M.D., MPH, Ph.D

Academic Editor

PLOS ONE

Additional Editor Comments (if provided):

I agree with Reviewer #1, that reviewer comments are to help you make the paper clearer for future reader. I also agree with they that the discussion need to be strengthen. Please use their comment to finalize the paper

Reviewers' comments:

Reviewer's Responses to Questions

**Comments to the Author**

1. If the authors have adequately addressed your comments raised in a previous round of review and you feel that this manuscript is now acceptable for publication, you may indicate that here to bypass the “Comments to the Author” section, enter your conflict of interest statement in the “Confidential to Editor” section, and submit your "Accept" recommendation.

Reviewer #1: (No Response)

Reviewer #2: (No Response)

2. Is the manuscript technically sound, and do the data support the conclusions?

Reviewer #1: Yes

Reviewer #2: Partly

3. Has the statistical analysis been performed appropriately and rigorously? 

Reviewer #1: I Don't Know

Reviewer #2: Yes

4. Have the authors made all data underlying the findings in their manuscript fully available?

Reviewer #1: Yes

Reviewer #2: Yes

5. Is the manuscript presented in an intelligible fashion and written in standard English?

Reviewer #1: Yes

Reviewer #2: Yes

6. Review Comments to the Author

Reviewer #1: GENERAL REMARKS

The authors have done a good job addressing the points raised in the review. There are however a few outstanding minor points and some comments from the first review that the authors are kindly requested to address.

1. Please add ‘of suspected TB’ to the following phrase “1 week before until 3 months after working diagnosis” where it appears in the manuscript

2. The authors provide the following explanation to the reviewer comments “The incidence density was calculated as (number of incident cases)/(person-time of subjects at risk in the unit of person-years).” However this definition/explanation should not only be for the benefit of the reviewer but should be included in the manuscript.

3. Page 7 Line 188: Please include the range in the following format “3 days-5 years” in the manuscript instead of 0.01-5

4. Page 8 Line 198: The authors indicate “There were no major differences in marital status, education, smoking behavior and alcohol consumption”. However the p-values for smoking, marital status are significant and there is no data for education in the Table. The authors should please address these discrepancies/gap.

5. It looks like the following points from the initial review were not addressed by the authors so they should please address them:

DISCUSSION

The authors indicate a high mortality of 14.3% among their EPTB patients. How does this finding compare with what other studies have found?

Given the various variables that the authors report on in their study, there is opportunity for rich discussion on the study findings. The authors should include more discussion on the poor outcome of their patients LTFU in view of the clinical characteristics reported on such as WHO stage, comorbidities and CD4 count.

In the conclusion, the authors highlight that lack of ART and age >45 years were risk factors for death or loss to follow-up. What recommendations do the authors offer for clinicians or program managers to address these findings?

Reviewer #2: Review report

PONE-D-19-26934R1

Extrapulmonary tuberculosis in HIV-infected patients in rural Tanzania: The prospective Kilombero and Ulanga antiretroviral cohort.

Major comments

ABSTRACT

2) You stated that: “The combined endpoint of death and LTFU was observed in 1058 (33.8%) patients, most frequently in the subgroup of EPTB (47.2%)”. You should give hazard ratio.

ANSWER: We constructed a cox model exclusively for patients with EPTB, so we could investigate risk factors for the combined endpoint in these patients. Therefore, no hazard ratios for comparisons of different study groups (e.g. EPTB vs. PTB) can be indicated, as the cox model did never include patients of e.g. the PTB group. However, with the data given in Table 3, the frequencies of the combined endpoint in the different study groups can be compared. We added details on the test used for the calculation of p-values in the table legend, in addition to the preexisting mention in the methods section.

GENERAL COMMENTS

I don’t agree. When you used cox-model you should compare groups of patients in the cohort. The comparison of the group with outcome to the one without outcome would give the hazard of the outcome e.g: HR should be specified.

As I stated, the major problem of the paper is the absence of methods to roll out other differential diagnosis (no cytological or histological examinations).

7. PLOS authors have the option to publish the peer review history of their article (what does this mean?). If published, this will include your full peer review and any attached files.

Reviewer #1: Yes: Dr Sally-Ann Ohene

Reviewer #2: Yes: Eric Walter PEFURA YONE

---

## [Author Response · Author response to Decision Letter 1]

5 Feb 2020

GENERAL REMARKS

Please find a for table S1 in line 122, page 5.

Reviewer #1: 

The authors have done a good job addressing the points raised in the review. There are however a few outstanding minor points and some comments from the first review that the authors are kindly requested to address.

1. Please add ‘of suspected TB’ to the following phrase “1 week before until 3 months after working diagnosis” where it appears in the manuscript.

 We added accordingly (Line 135 and 138).

2. The authors provide the following explanation to the reviewer comments “The incidence density was calculated as (number of incident cases)/(person-time of subjects at risk in the unit of person-years).” However, this definition/explanation should not only be for the benefit of the reviewer but should be included in the manuscript.

 We added this in the definitions section on page 6, line 164 ff.

3. Page 7 Line 188: Please include the range in the following format “3 days-5 years” in the manuscript instead of 0.01-5.

 We adjusted the manuscript accordingly on page 7, line 190.

4. Page 8 Line 198: The authors indicate “There were no major differences in marital status, education, smoking behavior and alcohol consumption”. However, the p-values for smoking, marital status are significant and there is no data for education in the Table. The authors should please address these discrepancies/gap.

We corrected as suggested: ‘Patients with EPTB were more frequently married and less frequently smokers.’ (page 8, line 199 and table 1). We deleted the information on education.

5. It looks like the following points from the initial review were not addressed by the authors so they should please address them:

DISCUSSION

The authors indicate a high mortality of 14.3% among their EPTB patients. How does this finding compare with what other studies have found?

 We added a section in the discussion, see page 14, line 302ff.

 

Given the various variables that the authors report on in their study, there is opportunity for rich discussion on the study findings. The authors should include more discussion on the poor outcome of their patients LTFU in view of the clinical characteristics reported on such as WHO stage, comorbidities and CD4 count.

The adjusted model did not show an association between indicators of HIV associated immunosuppression (WHO stage and CD4+ cell count) at baseline and the combined endpoint. This observation might be due to the inclusion of only patients developing EPTB into the model. By analyzing a selected group with a high probability of the endpoint as well as a presumed high level of functional immunodeficiency, WHO stage and CD4+ cell count might not further discriminate risk. Other parameters were not analyzed. We commented this on p 15, line 338 ff.

In the conclusion, the authors highlight that lack of ART and age >45 years were risk factors for death or loss to follow-up. What recommendations do the authors offer for clinicians or program managers to address these findings?

Lack of ART is an indicator of late presentation with insufficient time to start treatment in time, as ART initiation has to be postponed for 2-6 weeks in patients with EPTB not yet on ART. Therefore, the most important recommendation is early diagnosis of HIV and treatment start of ART. We adapted the conclusions on page 16, line 371 as follows: ‘The lack of ART as a factor for poor prognosis points at the importance of early HIV diagnosis and ART start’. 

We abstained from giving recommendations for elder patients with an age >45 years. Firstly, we consider it difficult to give a recommendation based on an observational study, secondly there might be other confounders such as other age-related diseases increasing the mortality. Please see page 16, line 367

 

Reviewer #2: 

ABSTRACT

You stated that: “The combined endpoint of death and LTFU was observed in 1058 (33.8%) patients, most frequently in the subgroup of EPTB (47.2%)”. You should give hazard ratio.

ANSWER: We constructed a cox model exclusively for patients with EPTB, so we could investigate risk factors for the combined endpoint in these patients. Therefore, no hazard ratios for comparisons of different study groups (e.g. EPTB vs. PTB) can be indicated, as the cox model did never include patients of e.g. the PTB group. However, with the data given in Table 3, the frequencies of the combined endpoint in the different study groups can be compared. We added details on the test used for the calculation of p-values in the table legend, in addition to the preexisting mention in the methods section.

GENERAL COMMENTS

I don’t agree. When you used cox-model you should compare groups of patients in the cohort. The comparison of the group with outcome to the one without outcome would give the hazard of the outcome e.g: HR should be specified.

As I stated, the major problem of the paper is the absence of methods to roll out other differential diagnosis (no cytological or histological examinations).

1) As your comment was pointing to results from table 3 where all study groups (i.e. including those with no TB) are summarized, we assumed you were also asking for HRs from a survival model of all patients in the study. As we explained in the previous comment and in the manuscript, we did not build a cox model including e.g. patients with no TB. This selection is based on the study question of finding risk factors in patients with EPTB. A model including also patients without any TB episodes would be much more complex (e.g. TB diagnosis would have to be modelled as a time-dependent covariate) and less interpretable without bringing any advantage for the answer of the study question.

However, we understand you comment as an indication for the need for a relative measure to compare the rate of the primary outcome in the patient groups with any TB as in Figure 3. To address this, we build a model including the PTB, EPTB and the EPTB/PTB groups and indicated HRs comparing those groups in the results section (p, line). This information replaces the results of the log-rank test in Figure 3 (added in the last round of review) in a more detailed way, so we removed it again.

In case this is not the information you were expecting, please specify in more detail the type of analysis to be conducted.

Abstract page 2, line 43. Results page 12, line 265.

 HR 95% CI p

PTB Reference 

EPTB 1.63 1.14-2.31 0.006

EPTB and PTB 1.65 1.15-2.36 0.006

2) We agree, that the absence of methods to roll out other differential diagnosis is a limitation of the study. We underline this in the discussion on page 16, line 349ff.

---

## [Editor Report · Decision Letter 2]

18 Feb 2020

Extrapulmonary tuberculosis in HIV-infected patients in rural Tanzania: The prospective Kilombero and Ulanga antiretroviral cohort.

PONE-D-19-26934R2

Dear Dr. Weisser,

We are pleased to inform you that your manuscript has been judged scientifically suitable for publication and will be formally accepted for publication once it complies with all outstanding technical requirements.

With kind regards,

Marcel Yotebieng, M.D., MPH, Ph.D

Academic Editor

PLOS ONE
---

## [Editor Report · Acceptance letter]

19 Feb 2020

PONE-D-19-26934R2 

Extrapulmonary tuberculosis in HIV-infected patients in rural Tanzania: The prospective Kilombero and Ulanga antiretroviral cohort. 

Dear Dr. Weisser:

I am pleased to inform you that your manuscript has been deemed suitable for publication in PLOS ONE. Congratulations! Your manuscript is now with our production department. 

With kind regards,

on behalf of

Dr. Marcel Yotebieng 

Academic Editor

PLOS ONE